# Leveraging Hot Spots of TEAD–Coregulator Interactions in the Design of Direct Small Molecule Protein-Protein Interaction Disruptors Targeting Hippo Pathway Signaling

**DOI:** 10.3390/ph16040583

**Published:** 2023-04-13

**Authors:** Bin Zhao, Ajaybabu V. Pobbati, Brian P. Rubin, Shaun Stauffer

**Affiliations:** 1Cleveland Clinic Center for Therapeutics Discovery, Lerner Research Institute, Cleveland Clinic Foundation, Cleveland, OH 44106, USA; 2Department of Molecular Medicine, Cleveland Clinic Lerner College of Medicine, Case Western Reserve University, Cleveland, OH 44195, USA; pobbata@ccf.org; 3Department of Cancer Biology, Lerner Research Institute, Cleveland Clinic Foundation, Cleveland, OH 44195, USA; rubinb2@ccf.org; 4Robert J. Tomsich Pathology and Laboratory Medicine Institute, Cleveland Clinic Foundation, Cleveland, OH 44195, USA

**Keywords:** TEAD, YAP, VGLL, protein-protein interaction, Hippo pathway, transcription

## Abstract

The Hippo signaling pathway is a highly conserved pathway that plays important roles in the regulation of cell proliferation and apoptosis. Transcription factors TEAD1-4 and transcriptional coregulators YAP/TAZ are the downstream effectors of the Hippo pathway and can modulate Hippo biology. Dysregulation of this pathway is implicated in tumorigenesis and acquired resistance to therapies. The emerging importance of YAP/TAZ-TEAD interaction in cancer development makes it a potential therapeutic target. In the past decade, disrupting YAP/TAZ-TEAD interaction as an effective approach for cancer treatment has achieved great progress. This approach followed a trajectory wherein peptidomimetic YAP–TEAD protein-protein interaction disruptors (PPIDs) were first designed, followed by the discovery of allosteric small molecule PPIDs, and currently, the development of direct small molecule PPIDs. YAP and TEAD form three interaction interfaces. Interfaces 2 and 3 are amenable for direct PPID design. One direct YAP–TEAD PPID (IAG933) that targets interface 3 has entered a clinical trial in 2021. However, in general, strategically designing effective small molecules PPIDs targeting TEAD interfaces 2 and 3 has been challenging compared with allosteric inhibitor development. This review focuses on the development of direct surface disruptors and discusses the challenges and opportunities for developing potent YAP/TAZ-TEAD inhibitors for the treatment of cancer.

## 1. Introduction

The evolutionarily conserved Hippo pathway is dysregulated in many diseases, including various cancers, and fibrotic and neurodegenerative diseases [1,2,3,4]. The Hippo pathway is formulated with upstream kinases and a downstream transcriptional complex comprising transcription factor TEAD (TEA/ATTS domain) proteins, the transcriptional coregulator YAP (Yes-associated protein), and its paralog TAZ (transcriptional coactivator with PDZ-binding motif) [5,6]. Recently, much has been learned about signaling mechanisms of the Hippo pathway [2,7,8]. This pathway is highly regulated by an upstream kinase cascade, including mammalian STE20-like protein kinase (MST1/2), which can phosphorylate and activate the large tumor suppressor (LATS1/2). The activated LATS kinases then phosphorylate YAP and TAZ, sequestering phosphorylated YAP/TAZ in the cytoplasm. Phosphorylated YAP and TAZ that remain in the cytosol of cells can be further ubiquitinated by an E3 ligase. Eventually, they are eliminated via proteasomal degradation. Thus, the concentrations of nuclear YAP and TAZ in normal cells can be regulated to an appropriately low level to avoid organ overgrowth. Conversely, when the kinase cascade is malfunctioning or genetically inactivated, unphosphorylated YAP and TAZ translocate into the nucleus. These transcriptional coregulators in the nucleus interact with TEAD proteins to induce downstream target gene expression which is pro-proliferative and anti-apoptotic, inducing the expression of genes such as *CTGF* (connective tissue growth factor) [9], *survivin* [10], and *CYR61* (cysteine-rich angiogenic protein 61) [11]. Indeed, high nuclear levels of transcription coregulators YAP/TAZ, achieved through both genetic and non-genetic mechanisms [12,13,14,15], are associated with the initiation and development of many human malignancies including lung, breast, pancreatic, prostate, colon, brain, and liver cancers and confer resistance to chemotherapeutic and targeted therapy drugs [16,17,18,19,20,21,22]. These observations suggest that dysregulation of YAP/TAZ–TEAD can result in uncontrolled tissue growth and cancer.

The four human TEAD family proteins (TEAD1-4) and their homologs in other species contain an N-terminal DNA-binding domain and a C-terminal transactivation domain (Figure 1A) where different coregulator proteins can bind. TEAD proteins alone do not have any transcriptional activity and therefore they recruit transcriptional coregulators via their C-terminal transactivation domain. On the other hand, coregulator proteins, including YAP and TAZ, need to bind to transcription factors, such as TEAD to modulate gene expression because they do not contain a DNA-binding domain. In addition to Hippo-dependent coregulator proteins, TEADs can also be regulated by vestigial-like family members (VGLL). There are four members of the VGLL family and VGLL1-3 have similar roles as YAP/TAZ in the tumor development [23]. However, studies have shown that VGLL4 is a natural YAP/TAZ inhibitor which can selectively antagonize YAP-dependent gene induction [24,25,26]. Given these data, obstructing the interaction of YAP/TAZ and TEAD has become an attractive strategy for a variety of YAP/TAZ-mediated TEAD dependent human cancers [27,28,29,30,31]. A TEAD inhibitor would be expected to be useful as a single agent against YAP/TAZ-mediated TEAD dependent cancers and could also be used in combination with other drugs where YAP/TAZ overexpression or activated TEAD is a major resistance factor [32].

As the field and biology have become better understood there have been increased anticancer drug discovery efforts and disclosures focused on designing TEAD inhibitors, particularly within the past decade [30,33,34]. Indeed, this has been demonstrated by the advancement of three TEAD inhibitors to reach phase I clinical trials, including VT3989 (Vivace Therapeutics, San Mateo, CA, USA), IK-930 (Ikena Oncology, Boston, MA, USA), and IAG933 (Novartis, Basel, Switzerland). The outcome of these ongoing clinical trials will be important to validate TEAD as a safe and efficient cancer target. Furthermore, given the wide implications of the Hippo signaling pathway in various biological and pathological processes, there is growing interest in evaluating TEAD modulators for other indications, such as heart disease, fibrosis, regenerative medicine and neurodegenerative diseases. For example, some research suggests TEAD proteins are involved in normal heart function and in the development of heart disease, determining cardiomyocyte and nonmyocyte proliferation and apoptosis [35]. In addition, TEAD proteins have been shown to play role in the development of fibrosis. These less understood indications are active areas of investigation in TEAD biology and the reader is directed to recent reviews that further describe developments in these areas [36,37,38,39]. Overall, TEAD modulators have shown promise as potential therapeutic agents for various diseases. However, further research is needed to determine their therapeutic potential and relative risks in humans.

This review focuses on the current status of direct YAP/TAZ-TEAD protein-protein interaction disruptors (PPIDs). We summarize the most recent progress made in the design of small molecule direct TEAD PPIDs that bind to the surface of TEAD and provide a perspective on the challenges and opportunities in modulating TEAD activity for the treatment of cancer.

## 2. Overview of TEAD Transactivation Domain Structures 

Since 2010, there have been 50 TEAD transactivation domain crystal structures deposited in the Protein Data Bank (PDB). They include apo, TEAD in complex with coregulator peptides (YAP, TAZ, mVGLL1, mVGLL4, FAM181A, and FAM181B), engineered peptides, TEAD mutants, and small molecules. These structural data have enriched our understanding of how TEAD interacts with different binding partners. In detail, the apo TEAD transactivation domain structure exhibits an immunoglobulin fold composed of two β sheets and two helix-turn-helix motifs (Figure 1B). In coregulator-TEAD complex structures one to three binding interfaces between coregulator proteins and TEAD are observed (Figure 2A–C). In addition, when overlaid with the apo and coregulator complex structures of TEAD, it became evident that the overall TEAD structure is almost identical and does not undergo major conformational changes upon binding to the coregulator proteins. In contrast, the TEAD binding motif of coregulators like YAP is intrinsically disordered in solution [40,41], but adopts a well-defined structure, including a β strand (interface 1), an α-helix (interface 2), and an Ω-loop (interface 3) (Figure 2A) when bound to TEAD [40,42,43]. The full-length human YAP protein comprises an N-terminal TEAD binding motif, followed by WW domain, coiled-coil, C-terminal activation domain, and PDZ motif (Figure 1A). The coregulators do not always form three interfaces after interacting with the surface of TEAD: YAP, TAZ, and VGLL2 wraps around TEAD and forms three binding interfaces; VGLL1, VGLL3, and VGLL4 interact with TEAD via interface 1 and 2 and lack an interface 3 interaction; the recently identified TEAD coregulator FAM181A binds to TEAD only at interface 3 [44], but its biological function is not clear yet (Figure 2D) [24,45]. However, interface 3 plays an important role in the high-affinity binding of YAP with TEAD.

Interface 1 is located at the surface of TEAD where an anti-parallel β sheet is formed between β strand 7 of TEAD and β strands of coregulator proteins YAP and VGLL. The C-terminus of the β strand of the coregulator protein is also near the highly conserved cysteine of TEAD, which is post-translationally autopalmitoylated [46]. The coregulator-TEAD crystal structures, including human YAP-TEAD and mouse VGLL1-TEAD complexes, suggest that the major interaction contribution of interface 1 is mediated by intermolecular hydrogen bonds between the peptide backbone ketone oxygen atoms and amide nitrogen atoms of coregulator protein β sheet residues and TEAD β7 residues. Early studies on YAP-TEAD interaction have suggested that the interface 1 β strand plays a marginal role in the overall YAP-TEAD binding affinity [42]. Several other studies have also confirmed that a truncated YAP peptide without the β strand lost less than two-fold binding affinity. This suggests that the interface 1 binding interface may have a minimal contribution to YAP-TEAD binding. This is not the case for the VGLL family proteins that bind to TEAD [45,47]. The mouse VGLL1 complex with TEAD4 structure showed that interface 1 possesses β1 and β2 strands. The β2 strand directly interacts with TEAD β7 and plays a key role in potency. Deletion of the β2 strand reduced mVGLL1 peptide binding to TEAD by almost 50-fold, as measured by TR-FRET assay and surface plasmon resonance [47]. Without the β strand, mouse VGLL4 and fly Vestigial (Vg) bound to TEAD and Scalloped (Sd) (Drosophila homolog of TEAD), respectively, with significantly weakened binding affinities [24,48]. Recently, Bokhovchuk, et al. reported that the interface 1 β strand of YAP showed 10-fold affinity difference between Sd and TEAD and was required for efficient binding to Sd but not TEAD [49]. The fact that the residues of the interface 1 β sheet in YAP/TAZ are less conserved, whereas those residues in VGLL proteins across species are well-conserved [50], is consistent with the observation that the β strand contributes more to the VGLL-TEAD interaction than to the YAP-TEAD interaction. Therefore, the interface 1 β strand plays different roles among co-regulators, such as YAP and VGLL proteins. Very few mutagenesis studies have been reported on interface 1 β sheet residues to explore the contribution of the side chains of residues to binding. Thus, one cannot exclude some of the side chains in the β sheet that may contribute to the binding interaction to some degree. For example, Jiao et al. replaced interface 1 β strand residues with poly-glycine in mouse VGLL4 and observed only minor binding changes [24]. The crystal structures revealed that the interface 1 β strands of coregulator proteins bind to TEAD in slightly different orientations, which may reposition backbone hydrogen bond angles or side chain interactions. This may explain why interface 1 β strands show varying degrees of contribution to the binding affinity. A detailed explanation of the observed molecular variability at interface 1 requires further investigation. Nevertheless, from a small-molecule drug discovery standpoint, the interface 1 region can be used to gain potency when synergistically combined with the interface 2-binding structural motifs.

Interface 2 is a hydrophobic groove formed by the helix-turn-helix motif (α helix 3 and 4) of TEAD. All structural data have shown that interface 2 is a highly conserved surface binding site for most TEAD co-regulator proteins, including YAP, TAZ, and VGLL1-4, with the exception of FAM181A (Figure 2A,B). A helix from the YAP, TAZ, and VGLL family packs into this binding groove and contacts TEAD using a very similar binding motif. The α helixes of YAP and TAZ contain a well-known hydrophobic binding module, LxxLF, which is located on one side of the α helix facing the binding groove of TEAD. We denote the hydrophobic pockets on the surface of TEAD occupied by the three residues of the LxxLF motif as P1, P2, and P3 (Figure 3). The α helix of the VGLL family contains a variant of this motif. It uses the VxxHF motif instead of LxxLF to form Interface 2. Although the interface 2 α helix from YAP, mVGLL1, and mVGLL4 binds to TEAD in a similar fashion to the same region on the TEAD surface, interestingly, the α helices of YAP and VGLL bind to TEAD in very different affinity ranges [51,52]. The isolated α helix of YAP showed an IC_50_ lower than 150 μM as measured by TR-FRET assay, but under the same experimental conditions, the α helix of mVGLL1 or interface 2 α helices of mVGLL4 (α2 and α3 helix) exhibited IC_50_ in a single-digit micromolar range [47]. Mesrouze et al. extensively explored the differences in the α helices of hYAP and mVGLL1 using structural data, site-directed mutagenesis, and other biochemical/biophysical approaches to understand the origin of their different affinities for TEAD [47,52]. The authors found that positioning of key residues, in particular Met40, His44, Arg47, and Ala48 helped mVGLL1 reach single-digit micromolar affinity. They also noticed that mVGLL1 and hYAP α helices bound to TEAD at different angles. This profoundly caused the residues at the N- and C-terminus of the α helix to occupy different binding positions to interact with TEAD. Subsequently, it also results in a different loop length connecting the β sheet and α helix. Altogether, these differences contribute to the potency gap between the α helices of hYAP and mVGLL1. The interface 2 α helix plays a key role in binding to all coregulator proteins, except FAM181A and FAM181B, which lack an α helix structural element. 

Interface 3 is referred to as the Ω-loop because of its twisted-coil topology, which resembles the Greek letter Ω. This interface includes the deep and broad surface pocket of TEAD that interacts with the key hydrophobic side chains of YAP residues Met86, Leu91, and Phe95. The center of this pocket is approximately 18 Å away from the C-terminus of the interface 2 α helix. Therefore, it is difficult to design small molecules that occupy both interface 2 and 3. In addition to these hydrophobic interactions, the residues from the Ω loop can also engage in hydrogen bonds and charge-charge interactions, such as Ser94^YAP^ with Glu263^TEAD4^ and Arg89^YAP^ with Asp272^TEAD4^. Interestingly, Phe96^YAP^ is located at the top of the pocket and does not directly interact with TEAD; however, the mutant Phe96Ala significantly reduces YAP binding by approximately 260-fold. This example shows that the Ω loop can be destabilized when the residue (Phe96^YAP^) that shields it from the solvent is mutated. Mutagenesis studies on the residues of the Ω loop region by several groups have confirmed that the interface 3 Ω loop is one of the two major contributors to the formation of the YAP-TEAD complex [40,42,43,53]. Overall, the binding specificity and affinity exhibited by YAP/TAZ and TEAD were mainly defined by hydrophobic and electrostatic interactions between the α helix and Ω loop of YAP/TAZ and the interface 2 and interface 3 regions of TEAD.

In 2015, five or six years after the first TEAD YBD domain crystal structure was reported, a hydrophobic central pocket was identified in the core of TEAD [51]. TEAD proteins can undergo autopalmitoylation via a highly conserved internal cysteine residue under physiological conditions and the palmitate occupies this central pocket [46,54]. Some studies have also demonstrated that palmitoylation might affect TEAD protein stability and, furthermore, might impact transcriptional regulation of the Hippo pathway [55]. This central lipid pocket is very hydrophobic, deep, and sufficiently large to bind small molecules. Clearly, this pocket is small molecule friendly, and druggable. This pocket is distinct from the YAP-TEAD surface interactions (interfaces 1–3) yet small molecules that bind to this pocket allosterically disrupt the YAP-TEAD interaction. For these reasons, most small molecule studies have focused on this allosteric central pocket to indirectly interfere with the YAP/TAZ-TEAD protein-protein interaction for anticancer drug development [28,29,30]. Since this disclosure, the field has rapidly evolved; with many drug development programs initiated in both academic institutes and pharmaceutical companies. Central pocket TEAD inhibitors have been reported to have significant TEAD inhibition both in vitro and in vivo, although the mechanism of action of these inhibitors remains unclear [29,30,31]. Some central pocket inhibitors which can inhibit TEAD target gene expression without necessarily disrupting the YAP/TAZ-TEAD interaction were observed as well [34,55,56,57]. Nevertheless, much progress has been made on this front, and two TEAD inhibitors, VT3989 (Vivace Therapeutics) and IK-930 (Ikena Oncology), that bind to the central pocket have entered clinical trials in 2020 and 2022 respectively. 

## 3. Strategies Employed in Targeting TEAD 

There are two major therapeutic approaches that interfere with YAP/TAZ-TEAD protein-protein interactions. According to the inhibitor-binding location, TEAD inhibitors can be divided into allosteric central pocket TEAD inhibitors and direct PPIDs that bind to the TEAD surface. An allosteric TEAD inhibitor binds to a highly conserved central pocket that houses the palmitate ligand and interferes with the palmitoylation process and/or protein stability of TEAD [55]. To date, many central pocket inhibitors have been reported. Although some of them do not disrupt the YAP-TEAD interaction and/or interfere with TEAD-mediated gene expression, several have achieved encouraging outcomes both in vitro and in vivo by targeting the YAP/TAZ-TEAD interaction, which we call allosteric PPIDs. Allosteric TEAD inhibitors have been reviewed in detail elsewhere [30,31,33] and will not be covered in this review. The other approach is directed at the contact area where the YAP/TAZ-TEAD proteins interact, aiming to directly block the YAP/TAZ-TEAD protein-protein interaction by binding the ligand at the TEAD surface. The major druggable interaction binding sites on the surface of TEAD are interfaces 2 and 3. 

### 3.1. Direct PPIDs Need to Target either Interface 2 or Interface 3 but Not Both

The TEAD-binding domains of YAP and TAZ contain a β strand (interface 1), α helix (interface 2), and twisted-coil region or Ω loop (interface 3). VGLL proteins, except for VGLL2, possess only one β strand and α helix. A YAP peptide with only the α helix was found to have a low affinity (>150 µM) for TEAD, while the VGLL peptides bind to TEAD via the same secondary structural elements with approximately a few hundred-fold higher binding affinity [47]. YAP and TAZ require an Ω loop to obtain sufficient affinity for TEAD. 

Furthermore, interface 2 (α helix) is a common binding interface among many TEAD co-regulator proteins (Figure 2D). This may reflect the importance of the binding mechanism between these proteins and the TEAD transcription factors. Detailed studies by Bokhovchuk et al. showed that the α-helix at interface 2 is formed before Ω-loop at interface 3 during YAP binding [48]. The author also pointed out that the α helix present in the intrinsically disordered proteins might facilitate the binding of coregulator proteins with TEAD because such secondary structural elements (α helices) often adopt a preorganized conformation in solution [58].

Intriguingly, the Ω loop of YAP^85–99^ bound to TEAD with an affinity of approximately 100 μM [44,51], while the α-helix bound to TEAD with less than 150 μM and a linker connecting interfaces 2 and 3 is necessary to have low nanomolar affinity between YAP and TEAD. This is reminiscent of the fragment-linking approach in which two weak fragments are linked to significantly improve binding affinity [59,60]. This additional increase in affinity can be explained when one considers that each of the fragments loses partial rotational and translational entropy and/or the linker contributes some affinity. However, it’s still very surprising to see such dramatic potency gain for the linked YAP peptide unless a significant contribution is derived from the linker. The crystal structure showed the amino acids from the linker region do not contact the TEAD protein. In addition, the linker region from other YAP homologs is very flexible and less conserved. All these data suggest that the linker or the amino acids from the linker region may not be the sole important contributor for the potency gain. Zhang and other researchers found residue V84 was at the C-terminus of the linker and indeed increased the binding a few fold when it was included in the Ω loop peptide [61]. Most likely, the role of V84 was to shield the Ω loop from solvent and maintain a hydrophobic environment. The V84 alone is not sufficient to explain the potency gain from a few hundred micromolar or even no detectable activity under certain experimental conditions to lower nanomolar affinity for the linked peptide. There must be some unique property of the linked peptide to explain why micromolar binding affinities of isolated the Ω loop and the α helix when connected by a flexible linker exhibit the nanomolar affinity. Tian et al. studied the TEAD-binding domain (TBD) of YAP using NMR and suggested that the N-terminus of the YAP TBD domain was an intrinsically disordered protein in its apo form [40]. It folds into a functional structure when it binds with a binding partner, such as the transcription factor TEAD. However, Feichtinger et al. showed another interesting feature of the apo YAP peptide [58]. They provided evidence that the apo state of YAP peptide in solution possesses a compact structure with distinct long-range correlations and also the Ω loop and the α helix structural elements are formed dynamically. This suggests the linked peptide in solution already has the correct binding mode and has overcome the induce-fit entropy energy penalty, and is ready to bind to TEAD. The role of the linker is not to simply connect the Ω loop and the α helix structural elements together, but may also keep the Ω loop and the α helix in close contact and/or initiate long-range communication between them. 

Nevertheless, the formation of a stable interface occurs through the two main contact regions, interface 2 (α helix) and interface 3 (Ω loop) between YAP and TEAD are essential for their binding interactions. These YAP/TAZ-TEAD interaction features also suggest that interrupting either binding site can significantly weaken the binding of YAP to TEAD. For an effective PPID design, targeting selected hotspot interactions is sufficient, and the entire interface does not need to be occupied by the small-molecule PPID. In addition, we anticipate that an effective PPID may not have to be very potent to efficiently block YAP/TAZ binding due to the relatively weak affinity of the individual interface 2 or interface 3 sites. Indeed, Novartis compound **4** only blocks interface 3 and has an IC_50_ of 150 nM and demonstrated efficacy in NCI-H2052 cells [62]. The TB2G1 peptide only blocks interface 2 to effectively disrupt YAP-TEAD interactions [63]. The feasibility of targeting only one of the two main sites to block the binding of YAP to TEAD has also been demonstrated by other peptidomimetic PPIDs, as discussed below.

### 3.2. Peptide-Based Small Molecule Inhibitors

Several studies have investigated the interaction between YAP/TAZ and TEAD using peptides and peptidomimetics. Most of the peptidomimetic PPIDs exhibited high affinity for TEAD, including cyclic YAP peptide cyclized through a disulfide bridge [61], a hybrid peptide from YAP and VGLL4 protein [24], TB1G2 which is a cysteine-dense peptide that mimics the interface 2 helix [63], and a linear YAP omega-loop peptide [64]. 

Peptide-based inhibitors are instrumental because they clearly show that either interface 2 or interface 3 sites can be leveraged for PPID design. A weakly binding 15-mer peptide corresponding to the Ω-loop sequence of YAP was transformed to bind TEAD with nanomolar biochemical potency. This was achieved by incorporating non-natural amino acids selected using a structure-based design [64]. This work demonstrated that it is possible to efficiently disrupt the YAP-TEAD interaction by targeting only the interface 3 Ω loop site.

The cystine-dense peptide TB1G2 showed that including a helical structure through the incorporation of interface 2 YAP helix residues in a cystine-dense protein scaffold can also generate an effective PPID that binds to the interface 2 region of TEAD with nanomolar potency [63]. Likewise, a cell-penetrating VGLL4 peptide is another example that shows that it is possible to efficiently disrupt VGLL-TEAD interactions by binding the interface 2 α helix site [65].

These peptide studies demonstrate that direct disruption of the YAP-TEAD interaction can be achieved by exploiting either interface 2 or interface 3 sites.

### 3.3. Direct Non-Peptide Small Molecule Inhibitors Targeting Interface 2 or 3

From peptide-based PPID studies, we may infer that direct targeting of YAP/TAZ-TEAD protein-protein interactions can be a straightforward approach to block YAP/TAZ binding with TEAD; however, this approach is challenging to achieve using non-peptide small-molecule inhibitors because there are large interaction surfaces (1900 Å^2^) between YAP/TAZ-TEAD interactions and there are only a few binding pockets for traditional small molecules to bind. The scarce number of reported surface-binding TEAD small-molecule TEAD PPIDs is a testament to the difficult road ahead (Table 1). Notably, a series of patents from Inventiva Pharma published between 2017 and 2020 showed that hydrazoboroaryl and benzo[d]isothiazole-dioxide derivatives can significantly block the interaction between YAP/TAZ and TEAD at interface 3 [66]. A cryptic pocket at interface 2 has also been reported [67]. The authors also determined the crystal structures, which showed that the cryptic pocket was created via the flipped side-chain phenol ring of Tyr382^TEAD2^ to accommodate the dichlorophenyl moiety present in a trisubstituted pyrazole scaffold. To identify the drug-like novel surface TEAD inhibitors, BY03 and CPD3.1, were identified through in silico screens [68,69]. Although they have been reported to disrupt TEAD-dependent transcription, additional structural information is required to fully understand the binding mode and mechanistic basis of these molecules.

Recently, Novartis reported a major breakthrough in the field and disclosed a very potent interface 3 small molecule YAP-TEAD inhibitor [62]. It is also the only surface inhibitor of YAP/TAZ-TEAD, named IAG933, that has entered clinical testing. The structure of IAG933 has not been disclosed, however, Novartis disclosed a patent describing this YAP-TEAD inhibitor scaffold in 2021. A linear YAP peptidomimetic designed by Furet et al. laid the groundwork for the design of this small molecule PPID [64]. First, the authors used a virtual screen to identify small molecule mimics of chlorotryptophane in the linear YAP peptide that bind deeply into the hydrophobic cavity at interface 3 of TEAD. The dihydrobenzofuran core was identified as a hit on the virtual screen. Chlorotryptophan mimics the TAZ W43 residue which is a hotspot for TAZ-TEAD interaction. This is a fine example of how to take advantage of hot spots and effectively reduce the need for small molecule occupancy over large, flat PPI interfaces to inhibit YAP-TEAD interaction. A structure-based rational drug design approach, coupled with medicinal chemistry efforts yielded compound **3**, which introduced a 4-phenyl substituent to the benzofuran core, boosting the binding affinity by two orders of magnitude (Figure 4). Ultimately, compound **3** was further developed into compound **6** to improve its binding affinity, and overall, the binding affinity of the initial hit was improved by five orders of magnitude using a structure-based design. Compound **6** exhibited a single-digit nanomolar IC_50_ value and inhibited YAP-TEAD complex formation as measured by TR-FRET. They also observed that this tight biochemical potency translated into very strong inhibition of YAP-TEAD-dependent gene expression in NCI-H2052 cells. 

Surprisingly, this dihydrobenzofuran fragment also coincidentally occupied interface 2 in the crystal structure (PDB: 8A0V) in addition to interface 3. The scaffold of the compound exhibits a V shape and kink in the middle so that the dihydrobenzofuran core is placed at the pocket as His44 from VGLL and the benzol ring mimics the highly conserved Phe69 of YAP to interact with TEAD. However, the compound was modeled in only two copies in the complex (chains A and B) in the reported structure. The authors did not model it at chains C and D, most likely because of either crystal packing or the low occupancy of the ligand. In addition, the average B-factor of the fragment at interface 2 was about 68 A^2^, which implies that the ligand could be more dynamic and not in a stable conformation. Nevertheless, these observations suggest that the fragment compound **1** could bind to interface 2, but its binding affinity to interface 2 is probably much weaker than that to interface 3. The exact binding affinity of fragment compound **1** to interface 2 remains unclear. The overall binding of both sites of this compound to TEAD4 was determined to be about 180 μM using 2D C13-N15 HMQC NMR spectra. 

## 4. Interface Hot-Spots That Can Be Tapped for Drug Discovery

Mutagenesis at interface 3 has been more extensively studied and highlighted the important role of these residues as hotspots in binding affinity (Table 2). For example, mutations were carried out in the TEAD-binding domain of YAP at interface 3 residues Met86^YAP^, Arg89^YAP^, Leu91^YAP^, Ser94^YAP^, and Phe95^YAP^ [70]. When mutated to alanine, all mutations significantly reduced the binding affinity of YAP-TEAD by more than 100-fold. To evaluate the impact of these key YAP interactions, TEAD mutations such as Glu263Ala, Val265Ala, Asp272Ala, and Trp299Ala were produced. Due to impaired interaction with YAP residues, all mutants showed lower binding activity than WT TEAD [40,42,70]. In addition, the hydrophobic side chains of Met86^YAP^, Leu91^YAP^, and Phe95^YAP^ point toward the center of the hydrophobic pocket and focus on the same hydrophobic area. They also interact with each other to stabilize bound Ω loop formation. This is a unique feature in that each of the three hydrophobic residues at interface 3 not only plays a crucial role in binding and is located in the same focused region, but also creates a stabilization network to form the binding mode. This is a centralized hotspot and provides a structural basis for small molecules to target the Ω loop site. 

The hot spots at interface 2 were identified and confirmed by sequence comparison, structural analysis, and mutagenesis [29,42,43,70]. The alignment of the amino acid sequences of the co-regulator proteins YAP, TAZ, and VGLL is presented in Figure 5. L^R1^xxL^R2^F^R3^ and V^R1^xxH^R2^F^R3^ motifs were labeled. R1 is Leu or Val among these coregulator protein members. At the R2 position, Leu is conserved in YAP and TAZ. However, among the VGLL 1–4 proteins, the R2 position is conserved as His. His is involved in polar interactions with Ser, which is fully conserved in TEAD members. The R3 residue is Phe, the only highly conserved residue across all co-regulator proteins possessing the α helix structural element, which indicates its importance in the interaction with TEAD. The fourth hydrophobic residue, R4 Val, is frequently observed in YAP and TAZ, whereas Leu is present in VGLL family members. Structural studies have shown that these four hydrophobic residues (R1, R2, R3, and R4) are located on one side of the α helix [40,45]. They are engaged in four hydrophobic binding pockets (labeled respectively as P1–P4, Figure 3A) in the interface 2 binding groove. Besides these prominent features, Arg and Lys are generally found at the second residue after the conserved Phe in VGLL family but these two residues are not present in YAP and TAZ proteins.

First, the majority of the mutation studies associated with coregulator proteins have been carried out to evaluate the relative importance of the three hydrophobic residues (R1–R3) in TEAD proteins [40,42,47]. For details on how these individual mutations reduce potency in comparison with wild-type YAP, TAZ, or VGLL peptides, refer to Table 2. Mutations in R1–R3 of YAP residues to Ala decreased the affinity to TEAD in all cases [47,52,70]. 

Mesrouze et al. measured the dissociation constant of the mutants using Surface Plasmon Resonance to quantitatively evaluate the contributions of key residues for binding [70]. At the R1 position, the mutant Leu65Ala^YAP^ lost its affinity 40-fold. In the VGLL family, R1 is valine; interestingly, the replacement Val41Leu^mVGLL1^ can cause a reduction in binding of about 6-fold [52]. Both Leu65^YAP^ and V41^mVGLL1^ are located at the C-terminus of the α helix, but the position of V41^mVGLL1^ is closer to the main chain of α4 helix of TEAD and does not allow a bulkier side chain at that position. Therefore, Val to Leu mutation will cause steric clashes with the α4 helix of TEAD. This observation is consistent with the mutagenesis results and suggests that the P1 pocket is more rigid and shallower.

At the R2 position, the mutant Leu68Ala^YAP^ lost affinity 25-fold. In addition, when comparing the YAP/TAZ and VGLL family members, the variation in the R2 position and the corresponding P2 pocket interaction was dramatically different. Leu68^YAP^/Leu15^TAZ^ occupies P2 and interacts primarily with Phe373^TEAD4^, Ser336^TEAD4^, Phe337^TEAD4^, and Val389^TEAD4^. However, His44^mVGLL1^ occupied the same P2 pocket across all VGLL members. The imidazole side chain of His44^mVGLL1^ formed hydrogen bonds with the backbone carbonyl oxygen of Val389^TEAD4^ and the hydroxyl group of Ser336^TEAD4^ [45]. To understand how this H-bond efficiently contributes to binding, Mesrouze et al. created a single mutation on hTEAD4. The residue Ser336^TEAD4^ was mutated to Ala [47]. Mutation of this specific Ser336 significantly affected the VGLL1:hTEAD4 interaction, as measured using the TR-FRET assay. VGLL peptide bound to the Ser336Ala^TEAD4^ mutant poorly, compared to WT TEAD, the binding affinity was compromised by about 30-fold. 

It is not surprising that the R3 to Ala mutant showed the most profound impact on affinity and reduced binding by approximately 400-fold. This was consistent with the sequence alignment analysis, which showed that R3 was the most conserved residue. 

At the R4 position, YAP binding with TEAD is tolerant to the mutation; for instance, the YAP R4 Val to Ala does not impact YAP-TEAD interaction. This indicates that the R4 residue is not essential for the binding affinity of TEAD for YAP. In addition, the complex structure showed that the P4 pocket is not well-defined and is more exposed to solvent compared to the other three pockets, which correlated with the lack of interaction of the residues in this pocket for YAP. However, Adihou reported that the mutant Leu242^mVGLL4^ significantly reduced mVGLL4 binding to hTEAD1 [65]. Leu242^mVGLL4^ is at the turn between α2 and α3 helix of mVGLL4 and occupies the P4 pocket. In addition, the TAZ complex structure showed that the R4 Val35^TAZ^ residue is on the P4 pocket, but it adopts different orientation and position compared with R4 Val of YAP and is more defined [71]. These observations indicated that the P4 pocket is more defined in the TAZ/VGLL complex than in the YAP complex. In addition, the R4 position in mVGLL1 is the residue Leu49^mVGLL1^ at the N-terminus of the α helix. The side chain of Leu49^mVGLL1^ forms more hydrophobic interactions with Leu373^mTEAD4^ and Met378^mTEAD4^. These results suggest that the P4 pocket is more plastic and broader than those of the other three pockets. 

On the other hand, mutants from the TEAD proteins also confirmed the importance of the role of these four hydrophobic pockets on the surface of TEAD [70]. For example, Lys376^TEAD4^ is located on the α3 helix and is formed on one side of the hydrophobic wall of the P3 pocket. The other side was exposed to solvent, which suggests it has a role in shielding the P3 pocket from solvent access. Mutation of Lys376 to Ala results in a more than 10-fold loss in binding affinity. Leu380^TEAD4^ is in the loop of helix-turn-helix near the P3 and P4 pockets within the distance of hydrophobic interaction with the highly conserved R3 residues. The mutant Leu380Ala^TEAD4^ can reduce binding by approximately 10-fold. Val389^TEAD4^ is located on the α4 helix and on the edge between P2 and P3 within the van der Waals interactions with both R2 and R3 residues. The mutant Val389Ala^TEAD4^ decreased binding by approximately 20-fold. These observations demonstrate that R1-R4 are key residues for the binding of TEAD. 

It is worth noting that although the isolated α helices from hVGLL1 and hYAP contain the conserved binding motif VxxHF or LxxLF, they did not show detectable binding affinity to hTEAD4 or a reported IC_50_ weaker than 150 μM as measured by TR-FRET assay [47,52]. This implies that the interaction from the three key residues (VxxHF and/or LxxLF) at the α helix is not sufficient to have single-digit micromolar potency. Additional contributions from other residues in this region must be considered as well. For example, Met40^mVGLL1^ can contribute to mVGLL1 binding to TEAD approximately seven-fold [47]. Arg47^mVGLL1^, conserved in the VGLL family as Arg or Lys at this position, also affects the potency of TEAD. Its side chain guanidinium moiety makes a cation-pi interaction with Phe337^TEAD4^, which is highly conserved in all TEAD proteins [52]. Besides the key residues, the orientation of α helix and the binding modes also dramatically impact the overall binding of interface 2 to TEAD.

Overall, interface 2 provides another opportunity for PPID design. A small molecule targeting this site must utilize several key contacts to improve potency. It should occupy the P3 pocket, engage at P2 in productive hydrogen bonds, potentially mimicking the His imidazole moiety, and preferably forge interactions similar to those of Met40 and Arg47 of mVgll1. As the key binding site on interface 2 is more dynamic and complex than the hotspot binding site on interface 3, the development of interface 2 inhibitors may be particularly challenging.

**Table 1 pharmaceuticals-16-00583-t001:** Selected non-peptide small molecule ligands that bind on TEAD surface pocket interfaces 2 and 3.

Molecule	Structure	Screening Method	Validation Method	Binding	Reference
Fragment 1	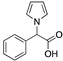	Thermal shift assay	Crystal structure	>300 μM; ITCInterface 2	[72]
BY03	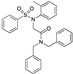	Molecular docking		9.4 μM; SPRInterface 2	[68]
Fragment compound **1**	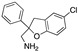	Molecular docking	Crystal structure	180 μM; NMR Interface 2 and 3	[62]
Trisubstituted pyrazoles	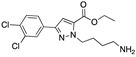	Pharmacophore hopping	Crystal structure	Interface 2	[67]
CPD 3.1	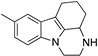	Molecular docking		12 μM, ITC Interface 3	[69]
Novartis Compound **4**	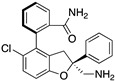	Molecular docking	Crystal structure	0.15 μM, TR-FRETInterface 3	[62]
InventivaCompound **4**	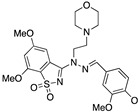	Fragment screening and HTS		Interface 3	[66]

**Table 2 pharmaceuticals-16-00583-t002:** Mutations of the residues of coregulator proteins and TEADs.

Protein	Mutation	Location	Pocket	Effect	Method	Phenotype	Ref.
hYAP	Leu65Ala	Interface 2	P1	Destabilizes YAP-TEAD complex (∆∆G 2.24 kcal/mol)	Surface Plasmon Resonance		[70]
hYAP	Leu68Ala	Interface 2	P2	Destabilizes YAP-TEAD complex (∆∆G 1.92 kcal/mol)	Surface Plasmon Resonance		[70]
hYAP	Phe69Ala	Interface 2	P3	Destabilizes YAP-TEAD complex (∆∆G 3.48 kcal/mol)	Surface Plasmon Resonance		[70]
mVgll1	Met40Val	Interface 2	other	TEAD4 interaction was reduced by 9-fold	TR-FRET		[47]
mVgll1	His68Leu	Interface 2	P2	Dramatic reduction in its ability to interact with TEAD	TR-FRET		[52]
mVgll1	Ala72Val	Interface 2	P4	TEAD interaction was reduced by 16-fold	TR-FRET		[52]
mVgll1	Arg71Ala	Interface 2	P2	Dramatic reduction in its ability to interact with TEAD	TR-FRET		[52]
mVgll4	Leu242A	Interface 2	P4	Dramatic reduction in its ability to interact with TEAD	Surface Plasmon Resonance		[65]
hTEAD4	Phe337Ala	Interface 2	P2	>90% reduction in the YAP-TEAD interaction; YAP interaction was reduced by 11-fold	Co-IP; Surface Plasmon Resonance	Decreased number of YAP/TEAD-driven soft agar colonies	[42,70]
hTEAD4	Ser336Ala	Interface 2	P2	Vgll1 interaction was reduced by more than 20-fold	TR-FRET		[47]
hTEAD4	Lys376Ala	Interface 2	P3	YAP interaction was reduced by 11-fold	Surface Plasmon Resonance		[70]
hTEAD4	Leu380Ala	Interface 2	P3, P4	YAP interaction was reduced by 9-fold	Surface Plasmon Resonance		[70]
hTEAD4	Val389Ala	Interface 2	P2, P3	YAP interaction was reduced by 17-fold	Surface Plasmon Resonance		[70]
hTEAD2	Glu267Arg	Interface 3		>90% reduction in the YAP-TEAD interaction	GST-pull down		[40]
hTEAD4	Glu263Ala	Interface 3		Destabilizes YAP-TEAD complex (∆∆G 1.19 kcal/mol)	Surface Plasmon Resonance		[73]
hTEAD2	Ile274Ala	Interface 3		>90% reduction in the YAP-TEAD interaction	GST-pull down		[40]
hTEAD1 hTEAD2 hTEAD4	Tyr421His Tyr442His Tyr429His			Destabilizes YAP/TAZ-TEAD complex (∆∆G > 2.9 kcal/mol)	Surface Plasmon Resonance		[74]
mTEAD1	Tyr410His	Interface 3		>90% reduction in the YAP-TEAD interaction	GST-pull down	Loss of transcriptional activity	[75]
hTEAD4	Tyr429Ala				Co-IP	Decreased number of YAP/TEAD-driven soft agar colonies	[42]
hTEAD2	Lys277Glu	Interface 3		>90% reduction in the YAP-TEAD interaction	GST-pull down		[40]
hTEAD2	Trp303Ala	Interface 3		>90% reduction in the YAP-TEAD interaction	GST-pull down	Decreased number of YAP/TEAD-driven soft agar colonies	[42]
hTEAD4	Trp299Ala				Co-IP		
hTEAD2	Leu444Ala	Interface 3		>90% reduction in the YAP-TEAD interaction	GST-pull down		[40]
hTEAD4	Lys297Ala	Interface 3		>90% reduction in the YAP-TEAD interaction	Co-IP	Decreased number of YAP/TEAD-driven soft agar colonies	[42]
hYAP	Met86Ala	Interface 3		TEAD4 interaction was reduced by 87-fold	TR-FRET		[70]
hYAP	Arg89Ala	Interface 3		Destabilizes YAP-TEAD complex (∆∆G 4.34 kcal/mol)	Surface Plasmon Resonance		[70]
hYAP	Phe95Ala	Interface 3		Destabilizes YAP-TEAD complex (∆∆G 4.31 kcal/mol)	Surface Plasmon Resonance		[70]
hYAP	Leu91Ala	Interface 3		Destabilizes YAP-TEAD complex (∆∆G 4.4 kcal/mol)	Surface Plasmon Resonance		[70]
hTAZ	Trp43Ala	Interface 3		TEAD4 interaction was reduced by 457-fold	TR-FRET		[53]
hTAZ	Lys46Ala	Interface 3		TEAD4 interaction was reduced by 29-fold	TR-FRET		[53]

## 5. Conclusions and Future Considerations

It remains challenging to develop potent small-molecule surface inhibitors to disrupt YAP/TAZ-TEAD interactions. The interaction interfaces are shallow and span a large surface area; however, significant advancements in the discovery of TEAD-YAP/TAZ PPIDs have been achieved over the past decade. This effort has resulted in a very potent interface 3 inhibitor (IAG933) as well as a number of allosteric inhibitors. The promising in vitro and in vivo preclinical biological activities demonstrates the feasibility of targeting the YAP/TAZ-TEAD complex using small-molecule inhibitors. In particular, IAG933 not only demonstrated the possibility of designing a non-peptide small molecule to target this broad, shallow PPI, but also provided strong motivation for the continued development of surface TEAD inhibitors. The TEAD-YAP/TAZ PPIDs at the Interface 2 site described to date are still at a very early stage in drug discovery. Significant challenges remain; however, with the recent increase in interest and accumulated knowledge, it is likely that these challenges will be overcome, and these efforts will lead collectively toward novel and high-quality TEAD-YAP/TAZ interface 2 and interface 3 inhibitors. These efforts will open up an exciting new arena for research on the Hippo pathway where validated PPIDs can be used to explore Hippo pathway biology.

## Figures and Tables

**Figure 1 pharmaceuticals-16-00583-f001:**
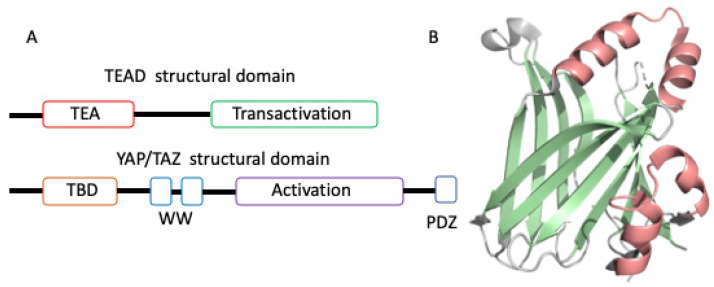
Structural domains of TEAD/YAP-TAZ and TEAD structure. (**A**) Domain architectures of TEAD and YAP/TAZ; (**B**) Ribbon diagram of the apo-TEAD crystal structure of TEAD2 (PDB: 5EMV). Two β sheets are colored in pale green and two helix-turn-helix motifs in salmon.

**Figure 2 pharmaceuticals-16-00583-f002:**
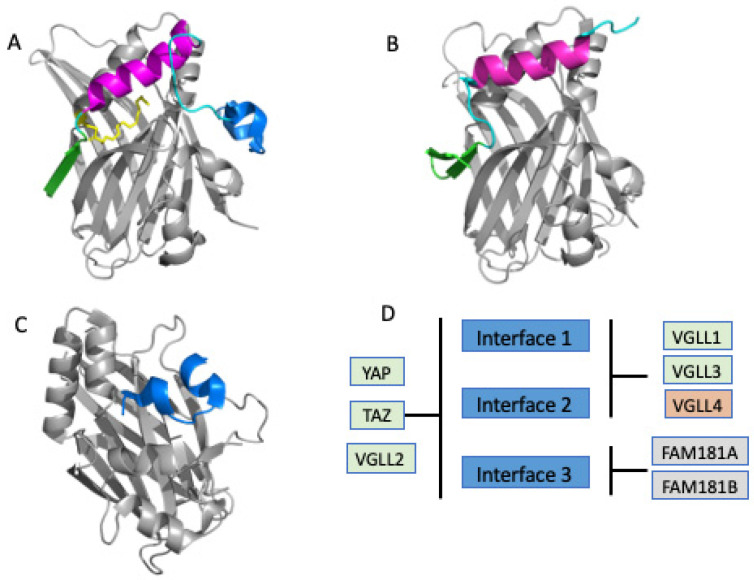
Ribbon diagram of the core structures of the complexes of TEAD-coregulators. (**A**) The YAP-TEAD complex (PDB: 3KYS); (**B**) the mVGLL1-TEAD complex (PDB: 5Z2Q); (**C**) the FAM181A-TEAD complex (PDB: 6SEN). The TEAD structure is displayed in grey, the interface 1, β sheet, in green; interface 2, α helix, in magenta; interface 3, the Ω loop, in blue; the loops in cyan; (**D**) Coregulators bind on different interfaces (light blue). Coactivator proteins are shown in light green, corepressor in orange, and proteins with unknown function in gray.

**Figure 3 pharmaceuticals-16-00583-f003:**
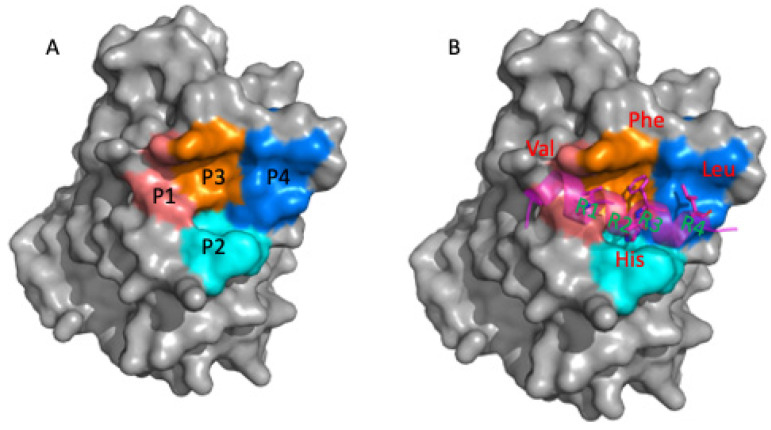
TEAD hotspots on interface 2. (**A**) Representation of the P1-P4 pockets in the binding groove of TEAD; (**B**) The mVGLL1-TEAD (PDB: 5Z2Q) complex. The interface 2, α-helix, peptide is represented in the transparent magenta ribbon. The mVGLL1 R1-R4 residues are shown as magenta sticks.

**Figure 4 pharmaceuticals-16-00583-f004:**
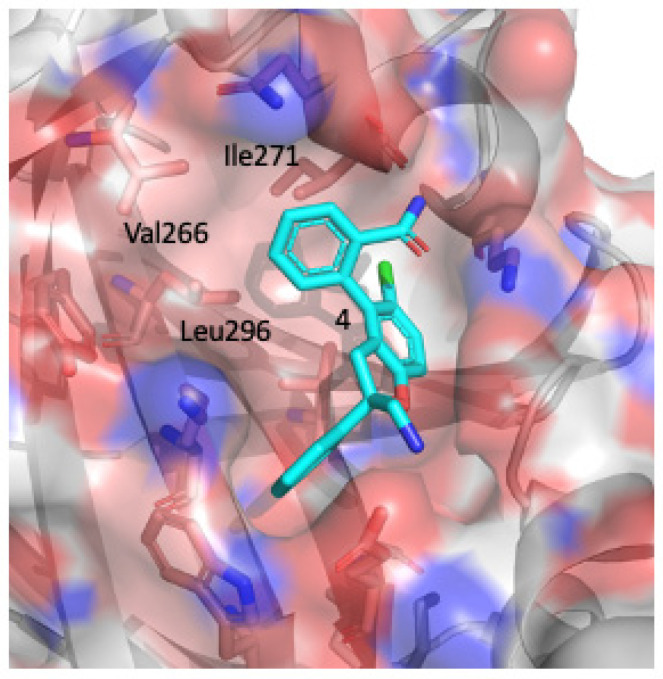
Binding mode of Novartis compound **4** in complex structure (PDB: 8A0U). The core structure is a dihydrobenzofurane ring with a phenyl substituent at position 4. The hydrophobic residues Ile 271, V266, and Lue296, shown in salmon colored sticks, interact with the phenyl substituent.

**Figure 5 pharmaceuticals-16-00583-f005:**
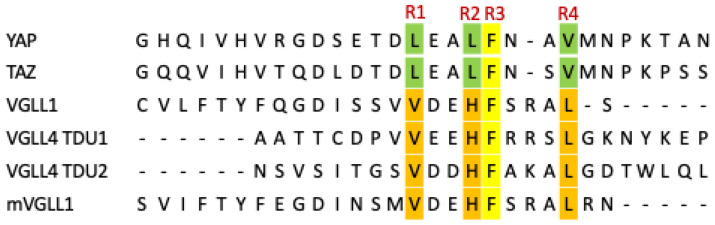
Sequence alignment of the interface 2 α-helix region of the coregulators. Four hydrophobic residues (R1, R2, R3, and R4) are labeled. The conserved residues in YAP/TAZ are highlighted in green; the conserved residues in VGLL family in orange; the highly conserved residues in all co-regulator proteins are highlighted in yellow.

## Data Availability

Data sharing not applicable.

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
