# Peer review of "Leveraging Hot Spots of TEAD–Coregulator Interactions in the Design of Direct Small Molecule Protein-Protein Interaction Disruptors Targeting Hippo Pathway Signaling"

_pharmaceuticals, 2023, doi:10.3390/ph16040583_

Round 1
Reviewer 1 Report
This is a very comprehensive review summarizing the most important and recent discoveries on YAP-TEAD Protein-protein interaction inhibitors. The article is ready to be accepted for publication after the following minor correction:
Figure 1A: please add PDZ motif in the graph.
Author Response
Point 1: “This is a very comprehensive review summarizing the most important and recent discoveries on YAP-TEAD Protein-protein interaction inhibitors. The article is ready to be accepted for publication after the following minor correction”
We appreciate very much the reviewer for acknowledging that this is a very comprehensive review summarizing the most important recent discoveries on YAP-TEAD PPIDs.
Point 2: “Figure 1A: please add PDZ motif in the graph.”
We agreed. The PDZ motif was added in Figure 1. The revised Figure 1 is on P13.
Reviewer 2 Report
This is a well-drafted manuscript systematically summarizing the current small molecules that target the Hippo transcription factors TEADs. The graph figures and tables are sufficient to illustrate the major points of the text, which are helpful and efficient in conveying the scientific knowledge to readers. Some suggestions were made for further improvement.
1. Figure 1A-D were mentioned in the figure legend, but only Fig. 1A and 1B were found in the provided manuscript.
2. Including a new section introducing the current application of the TEAD inhibitors in treating human cancers and other diseases would strengthen the manuscript.
Author Response
Point 1: “1. Figure 1A-D were mentioned in the figure legend, but only Fig. 1A and 1B were found in the provided manuscript.”
Yes. Figures 1A and 1B were in the manuscript. In addition, we have also changed Figure 2A-D in the manuscript. Figure 2A, 2B, and 2C is on P3 line 109. Figure 2A and 2B is on P4 line 166.
Point 2: “2. Including a new section introducing the current application of the TEAD inhibitors in treating human cancers and other diseases would strengthen the manuscript.”
We added and discussed the TEAD inhibitors/modulators’ applications for other diseases such as heart disease, fibrosis, regenerative medicine, and neurodegenerative diseases on P2 lines 83 to 94. We also included five recent reviews in the references, which have extensively covered this topic.